# A Survey of Emerging Memory in a Microcontroller Unit

**DOI:** 10.3390/mi15040488

**Published:** 2024-04-01

**Authors:** Longning Qi, Jinqi Fan, Hao Cai, Ze Fang

**Affiliations:** School of Integrated Circuits, Southeast University, Nanjing 210096, China; longn_qi@seu.edu.cn (L.Q.); 220236581@seu.edu.cn (J.F.); 220226267@seu.edu.cn (Z.F.)

**Keywords:** embedded NVM, emerging memories, FRAM, RRAM, MRAM, PCM

## Abstract

In the era of widespread edge computing, energy conservation modes like complete power shutdown are crucial for battery-powered devices, but they risk data loss in volatile memory. Energy autonomous systems, relying on ambient energy, face operational challenges due to power losses. Recent advancements in emerging nonvolatile memories (NVMs) like FRAM, RRAM, MRAM, and PCM offer mature solutions to sustain work progress with minimal energy overhead during outages. This paper thoroughly reviews utilizing emerging NVMs in microcontroller units (MCUs), comparing their key attributes to describe unique benefits and potential applications. Furthermore, we discuss the intricate details of NVM circuit design and NVM-driven compute-in-memory (CIM) architectures. In summary, integrating emerging NVMs into MCUs showcases promising prospects for next-generation applications such as Internet of Things and neural networks.

## 1. Introduction

In recent years, there has been an extraordinary proliferation in the popularity and adoption of edge computing, reaching unprecedented levels [1]. To prolong the longevity of battery-powered devices, energy conservation modes, such as complete power shutdown (normally-off), can be employed to minimize energy consumption during periods of inactivity [2]. Nevertheless, a significant drawback of this technology is the potential loss of data stored in volatile memory, which can result in substantial performance and responsiveness penalties [3]. An alternative approach is the use of energy autonomous systems, which are battery-less and depend on the energy harvested from ambient sources [4]. Due to the unpredictable nature of ambient sources and the limitations of energy harvesters, these systems often face operational challenges caused by frequent power losses [5]. Energy-efficient technologies are necessary to ensure the preservation of work progress in the event of a power failure, regardless of whether the power outage is intentional or unintended.

The conventional strategy to realize intermittent computing involves utilizing a nonvolatile memory (NVM) as a backup for on-chip volatile memory and processor states (flip-flops, latches, and registers) [6,7,8,9]. However, the process of sequential long-distance data movement between the volatile parts and the NVM requires many state transitions and significant overheads in terms of the execution time and energy [3,10]. To address this issue, emerging NVMs, such as ferroelectric random-access memory (FRAM), resistive random-access memory (RRAM), magnetoresistive random-access memory (MRAM), and phase-change memory (PCM), are incorporated into microcontroller units (MCUs).

Previous reports have proposed two types of FRAM, one based on ferroelectric capacitors (Fecaps) [11] and the other on ferroelectric transistors (FeFETs) [12]. A Fecap is a nonlinear capacitor with hysteretic behavior, forming the basis of nonvolatility. FeFETs exhibit nonvolatility due to the property of polarization retention in the absence of an externally applied electric field in ferroelectrics [13]. FeFETs may have a lower critical voltage for polarization switching compared to standalone Fecaps, provided they can be operated in the negative polarization-voltage region [14]. An RRAM device is based on a metal–insulator–metal structure and utilizes voltage pulses to create multi-level conductance levels, including high-resistance state (HRS) and low-resistance state (LRS) [15]. The magnetic tunnel junction (MTJ) device has a free magnetic layer and a pinned magnetic layer, which are separated by a thin insulator layer. It has a variable resistance depending on the direction of the write current [16]. PCM devices can modulate conductance based on the material phase, which can be switched by applying heat or voltage pulses [17]. These memory devices offer faster write times and lower write voltages than Flash memory [18], which makes it possible to provide instant on/off capabilities for MCUs with near-zero power consumption during the inactive phases [2].

Based on the emerging NVMs, nonvolatile static random-access memory (nvSRAM) is suggested as a replacement for the two-macro method (SRAM and Flash) because of its parallel operation and high speed [19]. The store operation of an nvSRAM bitcell involves, in nature, programming the nonvolatile elements according to the data held in the SRAM part. The active power consumption of nvSRAM has received attention and various strategies have been reported. For example, plate-line charge-share and bit-line non-precharge techniques were proposed to mitigate the active power of a 6T-4C nvSRAM [20]. Additionally, ensuring the reliability of real-time storage and restoration data in the nvSRAM is crucial for maintaining a successive workflow and service quality in MCUs. A bitcell-circuit-system co-design was employed to improve the reliability and scalability of a 64 KB RRAM-based nvSRAM, which was integrated into a 32-bit MCU and achieved a sub-0.1% raw bit error rate between the power outages [19]. Prior works have also investigated the possibility of utilizing the emerging NVM as an embedded Flash (eFlash) to enable rapid macro-to-macro backup and recall operations in the event of power loss [12,21,22]. In terms of reliable operations, a software-hardware programming configurable framework was adopted to achieve >95% read accuracy and a 1.3% mean error for all targets in a 3-bit mode of a 1 MB RRAM macro [21]. Despite numerous relevant studies, there is still a lack of universally applicable and significantly effective technologies for utilizing emerging NVMs as a replacement for SRAM and Flash in terms of power consumption, speed, and reliability.

Recently, state-of-the-art MCUs have supported machine learning (ML), allowing real-time data collection, ML model execution, and analysis on low-power devices. This advancement promotes the growth of computational intelligence at the edge by providing benefits such as improved security, privacy, reduced latency, and extended battery life [23,24,25,26]. Nevertheless, the pervasive deployment in edge applications is impeded by the substantial computational demands of ML inference, particularly in high-dimensionality matrix-vector multiplications (MVMs) [27]. To tackle this challenge, the concept of compute-in-memory (CIM) was introduced, which integrates high-efficiency computational logic within the memory array to significantly reduce memory and computation energy, thereby enabling the ML implementation on low-power MCUs [28,29,30]. SRAM-based CIM has gained increasing attention as a promising solution for ML applications [31,32,33,34,35,36,37,38,39]. Nonetheless, it exhibits inherent drawbacks including a high transistor count, limited data storage capacity, and unsuitability for long-term data retention, which makes it disadvantaged in terms of area, weight density, and event-driven applications [18,28]. In contrast, NVM-based CIM offers low standby power, high density with multi-level cells, and low system power by eliminating initial data writing [40,41]. It also efficiently stores weight-data for neural network models, with its capacity exceeding the mega-bit level. Despite these advantages, NVM-based CIM faces challenges due to memristor nonlinearity, and the need for large write currents and high-precision sense amplifiers, resulting in increased area and power consumption [18,30]. Therefore, this paper reviews existing design techniques for NVM-based CIM, aiming to inspire innovative circuit design strategies to address these challenges.

This article is a comprehensive review that will discuss the role of emerging nonvolatile storage in MCUs. The remainder of the paper is organized as follows. Section 2 discusses the feasibility of replacing SRAM and Flash in MCUs with emerging NVMs in terms of key parameters and the control scheme. Section 3 explores the design of NVM in an MCU, including the bitcell, read/write circuits, macro structure, and peripheral circuits. Section 4 reviews the design of CIM based on RRAM and MRAM. Conclusions are outlined in Section 5.

## 2. Feasibility of Replacing Flash and SRAM in MCUs with Emerging NVMs

### 2.1. Characteristics of Various Storage Types

Figure 1 depicts a prototypical MCU architecture comprising a processing core, Flash memory for code and data storage, SRAM for high-speed data access, and abundant peripheral devices such as PLL, DMA, UART, SPI, TIMER, WDT, GPIO, RTC, and PMU. The memory module primarily determines the power consumption of an MCU due to its intrinsic physical characteristics [42]. Therefore, selecting the appropriate memory for a low-power MCU is crucial.

MCUs integrated with Flash memory are commonly used in the commercial market. Due to its nonvolatile nature, Flash memory significantly reduces standby power consumption compared to volatile memories. However, the active operation of Flash memory requires high power consumption attributed to the need for high-voltage programming and erasing using charge pumps [42]. Moreover, integrating Flash memory into the advanced nodes has become increasingly complex and expensive due to its limited area shrink capability and growing complexity, as highlighted in several publications [43,44,45]. In response to these challenges, innovative eFlash designs were investigated, as detailed in Table 1 [46,47,48,49]. The SG-MONOS cell, combining split-gate and charge-trapping structures, enhances Flash memory performance and reliability by enabling efficient programming through source-side injection (SSI) and preventing column current leakage via series connection [46,50]. The eSTM is a floating gate-based cell, gathering the advantages of a conventional split-gate NVM cell together with a more compact cell area than a typical 1 T cell [49]. By using SG-MONOS and eSTM cell, eFlash macros were successfully fabricated at 40 nm and 28 nm with impressive specifications catering to high-end automotive applications. Nevertheless, these eFlash macros are constrained by low write endurance, making them difficult to use in applications that frequently power down. Furthermore, they tend to encounter elevated manufacturing costs attributable to the intricacy of the fabrication process, and persist in facing reliability challenges as the technology node continues to shrink.

Emerging NVM concepts (such as FRAM, MRAM, PCM, and RRAM) have been extensively researched to address these challenges. These alternatives offer easier integration into CMOS and lower process complexity [45]. Table 1 demonstrates the key features of type-like Flash NVMs based on emerging memory devices [22,51,52,53,54,55,56,57,58]. Type-like Flash NVMs refer to the emerging NVMs that directly store data in nonvolatile devices without the need for backup and restore operations and perform NOR Flash memory operations by imposing voltages corresponding to block-erase, random program, and random read. Previous reports indicate that FRAM consumes less power than Flash and DRAM, offering fast and high-bandwidth read/write operations [11,15,20]. Nevertheless, from the low clock frequency of FRAM-based MCUs in Table 1, it can be realized that FRAM still suffers from high power dissipation and limited clock frequency constraints when compared with other emerging NVMs. To remedy these constraints, a nonvolatile system-on-chip (NVSoC) integrated an instruction cache and increased the frequency to 30 MHz [10]. As indicated in Table 1, recent years have witnessed notable advancements in MCUs based on RRAM and MRAM, capable of reaching capacities in the megabyte range and working at lower operational voltages. A state-of-the-art MCU utilized four key design techniques to implement a 10.8 MB embedded STT-MRAM macro, which achieved the fastest random read access frequency and write throughput among reported Flash-replacement MRAMs [59]. However, the limited endurance of RRAM and reliability issues of MTJ impede their broader utilization [15,19,54]. PCM featured an attractive cell size of 0.019 F^2^ and attained the largest capacity of 21 MB among the embedded NVMs presented in Table 1. Nonetheless, PCM faces challenges due to the crystallization temperature limitations of the standard GST225 material, restricting its applications in consumer temperature ranges [45]. Recent developments have addressed this issue by using an optimized Ge-rich alloy with a higher crystallization temperature and a differential sensing scheme, enabling memory operations and data retention above 150 °C in a 32 KB embedded PCM [22]. In conclusion, each type of emerging NVM has its unique advantages and disadvantages, necessitating the selection of an appropriate storage type based on specific application requirements.

Conventional SRAM-based programmable logic has a large area and always consumes static power to maintain the stored data [60,61]. To reduce power consumption, the MCU enters the standby mode and employs multiple strategies to curtail the standby power, which is typically attributed to leakage current dissipation in the always-on domain. For instance, the cutting-edge ultra-low leakage MCU, leveraging 55 nm TFET-CMOS hybrid technology, achieved significant standby power reduction through innovative TFET-Gated-Ground SRAM and voltage-stacking techniques [62]. However, switching between active and standby modes consumes additional power for data transfer between volatile and nonvolatile memories [19,42,63]. Therefore, nvSRAM was proposed as a replacement for the traditional two-macro approach due to its high-speed parallel operation. The NVM elements and SRAM part are integrated in an nvSRAM bitcell by a bit-to-bit connection instead of a macro-to-macro connection [19,64]. By storing commonly used routines in the nvSRAM, the startup latency and associated energy consumption from data movement can be eliminated, enabling more efficient edge computing applications [65].

A comparison is performed in terms of several key parameters, shown in Table 2, to highlight the performance characteristics of MCUs with volatile SRAM and nvSRAM based on emerging memories [19,20,64,66,67,68,69,70]. Conventional SRAM consumes a much higher retention current than the deep-sleep-mode current required by sensor networks and energy-management systems, particularly as process geometry scales down. This has prompted investigations into leveraging low-leakage transistors and advanced design methodologies to achieve lower standby power levels. By utilizing thick-gate-oxide transistors with source bias control techniques, the retention power of the system was effectively reduced to the order of nanowatts [66,68]. However, these transistors generally cause an increase in memory macro area and active power dissipation, requiring additional techniques for minimizing cell size and active energy. An ultra-low-voltage MCU featuring single-rail SRAM was developed using 22 nm FDX technology and an adaptive reverse body bias scheme, achieving a leakage power of 6.6 μW and active power of 6.3 μW/MHz [67]. Nevertheless, the single rail macro incurs a 20% area overhead compared to the dual rail macro, and a custom bitcell design is needed to ensure stable read operations down to 0.5 V, thereby amplifying the complexity of the design.

In contrast, nvSRAM can be completely powered off during idle periods, thereby eliminating retention power consumption. The 4T-2MTJ macro, as illustrated in Table 2, exhibits the potential to achieve a smaller footprint compared to 6T SRAM at the 45 nm technology node, while maintaining an almost unchanged operation current over generations compared to the exponentially increasing current of 6T SRAM due to MOSFET off-current degradation with scaling [71]. In addition, innovative plate-line charge-share and bit-line non-precharge techniques effectively mitigate the active power dissipation from the large Fecap, making the 6T-4C macro suitable for an electrocardiograph monitoring SoC [20]. The 12T-2R nvSRAM showcases a raw restore-BER of less than 0.1% between power outages, contributing to the realization of an error-free nvSRAM macro with correction techniques [19]. One drawback of this nvSRAM is the 123% increase in area overhead compared to conventional 6T SRAM implemented on the same technology node.

### 2.2. Peripheral Circuits of Flash and SRAM

The Flash memory and SRAM exhibit distinct control principles based on their specific functionalities and operational requirements. A typical eFlash system is presented in Figure 2a and can be structured into three levels, including memory cells, peripheral circuits of the hard macro (such as sense amplifiers and high-voltage generators), and various functional blocks on a system level. The design of memory cells and peripheral circuits plays a vital role in determining the electrical characteristics and data reliability, and achieving the target specifications of the Flash memory. To realize higher performance and reliability in Flash memory, more CG or SL stitch regions may be required for faster rise/fall time or noise suppression. Furthermore, finer array division and control are needed to suppress the influence of program disturb on unselected cells, which is caused by sharing nodes among cells during program and erase operations.

On the other hand, the peripheral circuits of a typical SRAM consist of the timing control circuit, the address decoder circuits, the row (X) and column (Y) driver circuits, the sense amplifier circuit, the data input and output circuits, and the memory controller circuit. The timing controller synchronizes signal timing for proper operations of all SRAM components. The X/Y decoders select specific memory cells, drivers provide necessary voltage levels, and sense amplifiers ensure data integrity during read operations. The memory controller is a critical component, which manages data flow, controls read/write operations, and coordinates transfers between the CPU and SRAM. In SRAM design, ensuring stability across varying temperature and process conditions presents a significant challenge. A proposed SRAM design tackles this challenge by utilizing charge sharing to transfer stored charge from local bit lines (BLs) to global BLs, thereby ensuring a constant charging current for the BLs [66].

## 3. Design Considerations for NVM in MCU: A Focus on Three Metrics

### 3.1. Bitcell Design: Cell Size Focus

Figure 3a,c,e illustrate the classical configurations of nvSRAM bitcells, which integrate a traditional 6T-SRAM cell with nonvolatile elements and additional access transistors. When continuously powered, the nvSRAM bitcells function equivalently to standard 6T SRAM cells, storing data in cross-coupled inverters with comparable read and write speeds to traditional SRAM. This operational state is commonly called “SRAM mode” or “normal mode”. While encountering a power outage, these cells capture the SRAM contents in nonvolatile devices just before power loss and then restore the saved state when receiving power again.

The 6T-4C FRAM bitcell shown in Figure 3a is based on the Fecap, which is a nonlinear capacitor with hysteretic behavior. It has a much higher signal margin than the regular FRAM bitcells which utilize a single ended Fecap and two differential Fecaps, respectively [11]. This is because data are stored in all four capacitors in a complementary fashion, which also causes a higher area and power cost. Figure 3c shows the structure of the nvSRAM bitcell based on the SHE-MTJ, which offers significant advantages over traditional two-terminal MTJ [64]. Conventional MTJs are challenged by high switching currents and the need to balance resistance levels for read/write operations, compromising either writability or read sensitivity [80]. In contrast, the SHE-MTJ capitalizes on the SHE effect for enhanced spin generation efficiency and offers a low-resistance write path through charge current in the SHE-metal. Moreover, the decoupled read and write terminals facilitate the independent fine-tuning of the MTJ and SHE-metal dimensions, optimizing both readability and writability [64]. In the nvSRAM cell, M7 and M8 serve as extra access transistors that are disabled to separate the SHE-MTJs from the standard 6T SRAM cell for regular SRAM operation, and are activated to realize store and restore operations in the event of a power loss. This approach helps to prevent unnecessary write operations on SHE-MTJs when the data stored in the cross-coupled inverters change during continuous power on, thereby improving energy efficiency. The bitcell depicted in Figure 3e adds four clock-controlled power-gating transistors (M9~M12) to save energy, and it adopts the two-ends nvSRAM scheme [81,82,83] with two RRAM devices in a bitcell to improve reliability in low-HRS/LRS scenarios through differential sensing [19]. Despite its larger area and energy consumption compared to single-end nvSRAM cells [84,85,86], this configuration offers a superior sense margin and restoration yield for emerging NVM technology. It is possible to simplify the 12T-2R bitcell by replacing the M9, M11 pair and M10, M12 pair with a single transistor, respectively, but it is less efficient than realizing a compact transistor pair through area-sharing. The 12T-2R bitcell has an area overhead of approximately 123% when compared to a conventional sideway 6T SRAM bitcell at 130 nm [19]. Despite the ability of nvSRAM to combine the fast read/write characteristics of SRAM with the non-volatility of NVM, the cell structure based on 6T SRAM limits the reduction in cell size, resulting in poor scalability. When the extra area overhead outweighs the energy benefits it brings, the significance of this design becomes less evident. Therefore, nvSRAM struggles to achieve high-capacity storage and cannot effectively replace traditional embedded NVM in MCUs.

Figure 3b,d,f,g illustrate the standard type-like Flash bitcells, in which data are directly written into nonvolatile elements to eliminate the requirement for data backup and restore operations across power losses. Figure 3b displays a 1FeFET bitcell that leverages FeFET polarization for storing data and performs NOR flash memory operations in a NOR-type array. Two architectures for 1FeFET-based NOR Flash were introduced in [73]. One architecture offers a high level of scalability, achieving 6 F^2^ at a minimum by sharing the source lines in pairs of rows, while the other features more isolated cells, resulting in reduced disturbance but at the expense of scalability. A 1T-1FeFET bitcell with separate read and write paths was reported in [87], achieving non-destructive read and lower write power at iso-write speed compared to 1FeFET FeRAM. However, a slight area penalty is introduced due to the additional MOSFET. Furthermore, a 2T-1FeFET bitcell with separate read/write paths was designed to enhance design flexibility for CIM. Although the 1FeFET/1T-1FeFET bitcells are more compact, they require additional bias circuitry and/or charging of all non-selected WLs and BLs [75,88]. These introduce energy penalties and design complexities owing to the need for multiple voltage levels, rendering them less ideal for intermittently powered systems [12].

Figure 3d,f,g demonstrate a similar bitcell structure consisting of a MOSFET as an access transistor and a nonvolatile device as a storage element. Emerging NVMs based on this bitcell structure commonly serve as a replacement for embedded NOR Flash and show write speed and energy advantages over NOR Flash. The 1T-1MTJ bitcell, shown in Figure 3d, was reported and a local source line (SL) array scheme was implemented to improve write performance [77]. This scheme utilizes a local SL to distribute return current among unselected BLs for preventing select transistor TDDB stress, and it ensures no disturbance occurs in unselected BLs during write operations by connecting a group of MTJs to the local SL. It also enables the concurrent writing of 0 or 1 states without needing to elevate BLs, enhancing write efficiency in MRAM. Figure 3f illustrates a 1T-1R bitcell occupying an area of 20 F^2^ [54], which is more compact than earlier FRAM [89,90], STT-MRAM [16,76], and CBRAM [91,92] designs. The RRAM was equipped with a novel sense amplifier and a write-and-verify (WAV) voltage generator to increase the read and write yield. The 1T-1PCM cell, shown in Figure 3g, is fabricated by 0.11 μm BCD technology and covers an area of 0.7 F^2^. It utilizes a Ge-rich alloy for a higher crystallization temperature [93] compared to the conventional Ge_2_Sb_2_Te_5_ alloy, and it adopts differential sensing to mitigate resistance drift [94,95,96]. These features enable reliable memory operations and data retention at temperatures above 150 °C [22].

### 3.2. Read/Write Circuit Design: Power Efficiency Focus

As for read schemes for NVM, there are two typical sense amplifiers (SAs): the voltage-mode SA (VSA) and the current-mode SA (CSA). The VSA is used for precharging selected BLs to a target voltage, allowing the reading of both LRS and HRS cells. However, the limited voltage difference between HRS and LRS cells makes it susceptible to BL noise and coupling [97]. The CSA imposes a fixed bias voltage on the BL to induce current in the cell for reading. A current comparator is used to compare the sensed current with a reference current. Compared with VSA, the CSA minimizes the vulnerability to BL noise and coupling. Moreover, it exhibits faster read speeds than VSA when the BL length of a 0.18 μm RRAM macro exceeds 128 rows [97].

The read scheme proposed in [54] uses two types of CSAs to limit the read voltage to 0.3 V and avoid read disturbance, as shown in Figure 4a. By detecting the current of a RRAM cell at 0.3 V, unwanted state transitions, especially in the HRS, can be prevented. Current mirrors are used to provide magnifying power modes and compare the transformed voltage with the reference voltage (VREF) to determine the logic output. A modified VSA was presented for single-ended FeFET NVM, with duplicated read-BL voltages across two cross-coupled, inverter-based SAs connected to VREF-NAND and VREF-NOR during sampling [12]. During read operations, the SA with VREF = 0.95 V is enabled to sense the stored bit, with the inverted NOR output serving as the final READ (or OR) output. A fully configurable offset-tolerant CSA was reported previously [21], which can be tuned by software for precharging, calibrating, and latching. The design includes a dual-mode reference generator that can switch between a compact current-mirror-based mode and an accurate resistance network-based mode, providing a wider range of reference currents for multi-bit programming and read requirements.

The design of write peripheral circuits plays an important role in determining the write error rate and power consumption of NVM, particularly in the case of STT-MRAM. The challenges in write operations arise from the need for both a sufficiently long write time and a high write voltage to avoid errors [98]. However, the exact write time for each STT-MRAM cell differs due to process variations and thermal fluctuations. To ensure reliable writing, the write time is typically kept much longer than the average write time, causing energy wastage as the write current continues to flow even after the MTJ has switched [99].

The enhanced current programming circuitry depicted in Figure 4b improves RESET pulse shaping in high-parallelism programming scenarios by introducing a fast recovery method [22]. By incorporating an additional branch with transistor PP, the circuit ensures the rapid discharge of node A during programming pulses, allowing precise control of the current flowing through transistor P1. This approach eliminates the need for the precise matching of NMOS transistors and enables higher parallelism without increasing static power consumption or area occupation. Additionally, adjusting the values of factors α and β determines the circuit speed and the timing of turning off transistor P1, offering flexibility and efficiency in current programming operations.

Furthermore, the write termination scheme has been extensively discussed in previous research as an effective method for addressing the issue of energy waste during write operations. In this context, the energy waste arises from unnecessary write operations, which occur when the incoming data equal the current value stored in the memory cell. The self-write-termination (SWT) circuit monitors the write operation and prevents redundant writes, thereby improving energy efficiency and reliability in fast-switching cells. As reported in [99], the implementation of an SWT circuit leads to a remarkable 83% reduction in total write energy compared to conventional write circuits. Additionally, this approach utilizes 75% fewer transistors than previously proposed SWT circuits. In another study [52], the integration of SWT into each column of a 1-macro nvSRAM array demonstrates improved clock frequency and substantial reductions in store energy, up to 172×. Although considerable progress has been made, challenges remain in mitigating the extra power consumption and large area overhead associated with the SWT circuits.

### 3.3. Macro Structure and Peripheral Circuit Design: Area Efficiency Focus

The memory controller serves as a crucial component in managing data transfer between the CPU and memory modules in an MCU. It controls operations such as the reading, writing, and refreshing of memory modules to ensure that data are correctly stored and retrieved. In response to increasing demands for higher speed, lower power consumption, and an enhanced reliability of memory in various applications, previous studies have reported diverse memory controllers which integrate additional circuit modules to support functionalities beyond basic operations, as shown in Figure 5.

In recent years, MCUs with low-power and instant-on features have been highly valued for energy harvesting as well as “normally off” applications. Unfortunately, the conventional data backup strategy tends to store all contents from volatile parts into the NVM, even though most of the data are rarely changed or utilized in practical scenarios. To reduce unnecessary backup operations, a space domain controller was designed, as shown in Figure 5c, to provide the proper store address range of the nvSRAM [101]. The SWT circuit in each column detects bit changes and controls the write driver on the RSL line to terminate the SET or RESET operations as required. This approach eliminates the need to redundantly store or restore unused data, leading to reduced time and energy consumption during storage and restoration processes. A similar strategy was also adopted in [21], as depicted in Figure 5a. The memory controller supports the pre-read function that allows the controller to assess the resistance range of the RRAM device before writing to reduce redundant write operations. For restore operations, an adaptive parallel controller (Figure 5c) was employed to manage different restore parallelism options (1 WL/4 WL/16 WL) based on the maximum tolerant peak current of the power source and to restore the speed requirement, which contributes to the realization of the instant-on operation of the MCU. It is also common to utilize a memory controller for the power management of memory macros. As shown in Figure 5b, the memory controller deactivates the power gates of unselected RRAM modules, and it fully powers down the RRAM after a sufficient period following the last event, leading to an 89.21% reduction in power usage [100].

In memory design, reliability is crucial for ensuring data integrity and system stability, particularly in aerospace systems, medical devices, and autonomous vehicles. As illustrated in Figure 5d, a custom RRAM controller enhances yield to approximately 100% by integrating features such as built-in self-test, built-in self-repair, a shortened Bose–Chaudhuri–Hocquenghem (BCH) error-correlating code (ECC), and asymmetric coding [54]. Significantly, it enables adaptive ECC algorithm selection (Hamming or BCH code), resulting in improved performance with reduced power consumption, minimized parity bit overhead, and increased operation speed. The memory controller shown in Figure 5a is also equipped with an ECC module (ECC encoder and ECC decoder) to rectify erroneous bits utilizing the BCH algorithm [21].

The voltage generator is also a critical component in the memory macro, providing essential and precise voltage references for various operations. For example, two types of voltage generators, illustrated in Figure 5a, serve different purposes. The on-chip reference and voltage generator supplies constant and temperature-compensated references through the bandgap circuit, adjusting to accommodate diverse programming and readout needs [21]. The integrated charge pump produces a high voltage for the forming/set operation of the ReRAM. Additionally, a dual-mode reference voltage generator was designed to meet the diverse requirements of multi-bit storage memory. The current-mirror mode offers a wide range of reference currents for multi-bit programming needs, while the resistance network mode supports high/low-resistance states with high linearity, ensuring accurate voltage references for programming and readout. The integration of these circuits eliminates the need for off-chip references or high voltages, greatly simplifying the system design and connectivity. Furthermore, the design of the voltage generator can also enhance the reliability of memory. As illustrated in Figure 5d, the WAV voltage generator produces eight stepwise voltage levels in the BL or SL path, aiming to enhance write yield by mitigating variations in the transition energy of individual cells [54].

## 4. Circuit Design for CIM Based on RRAM and MRAM

Among emerging NVMs, RRAM and MRAM are the primary choices for embedded nonvolatile CIM due to their advantageous characteristics. RRAM exhibits a relatively larger on/off ratio than MRAM, less power consumption than PCM, and a higher compatibility with CMOS process than FeFET [30,102]. Moreover, its efficient performance in MVM operation with a crossbar structure has aroused extensive attention as a promising candidate to implement embedded NVM-based CIM [41,87,88,89,90,91,92,93,94,95]. On the other hand, spintronic devices provide a superior solution for nonvolatile logic-in-memory (LIM) architecture, enabling the efficient integration of a broad memory bandwidth in logic circuits [103,104]. STT-MRAM, as a representative spintronic memory, stands out for its lower access latency, superior endurance, and better process variation control compared to RRAM and PCM, making it well suited for embedded CIM [103,105,106]. Two silicon-validated examples of CIM will be reviewed later to explore the considerations in CIM design based on RRAM and MRAM technology.

Traditional RRAM-based CIM designs face two major issues in energy harvesting systems [107]: (1) the digital-to-analog (D/A) and analog-to-digital (A/D) conversion circuits between the RRAM array and the CPU significantly reduce energy efficiency and increase chip size; (2) all access transistors have to be turned on during each MVM operation, resulting in high sneak currents and unnecessary energy consumption. To overcome the existing limitations, a redesigned low-power MVM engine (Figure 6) has been introduced, which incorporates a binary interface and input-controlled access transistors [107]. By incorporating the binary interface, a direct link is established between the binary input vector and the WLs, with outputs obtained through the 1-to-3-bit adaptive SAs at the end of the BLs. This cuts out the A/D and D/A overheads, saving 44% in energy and 95% in area. Moreover, a 64% energy reduction is achieved by keeping access transistors off when inputs are zero. The proposed structure provides notable benefits particularly for networks with binary weights and input/output and contributes to the development of a smart processor that attains an energy efficiency of 462 GOPs/J at a clock frequency of 20 MHz.

Previous research has mainly focused on small-scale primitive logic-circuit elements, memory-like structures like FPGAs and filters, or simulation-based assessments, due to the lack of a well-defined design process tailored to MTJ technology in the chip fabrication environment. From this point of view, design flows for MTJ/MOS-hybrid logic circuits have been presented to realize practical-scale logic LSI based on a nonvolatile LIM architecture [103,108]. Utilizing the design flow, Figure 7 showcases a compact MTJ-based full adder (FA) with nonvolatile LIM architecture, enabling efficient, fully parallel processing for high-speed motion-vector extraction. The proposed FA exhibits a dynamic power consumption of 16.3 μW at 500 MHz, significantly lower than CMOS-only-based FA designs. A motion vector prediction unit was developed, comprising twenty-five processing elements (PEs) equipped with the reported FAs. It maintains intermediate data in nonvolatile memory, enabling precise power control during each operation cycle, which further diminishes leakage power and total power consumption [61].

Table 3 shows a comparison of state-of-the-art CIM designs based on volatile SRAM [36,38,39,109] and emerging nonvolatile memories [100,110,111,112,113,114]. For SRAM-based CIM design, various techniques were proposed to efficiently perform multiply-accumulate (MAC) operations, such as bit-serial multiplication and parallel adder trees [109], a segmented-BL charge-sharing scheme [38], and a time-domain incremental-accumulation scheme [39]. Additionally, innovative circuit structures and schemes, including 6T local-computing cells [36], source-injection local multiplication cells, prioritized-hybrid-ADC [38], and dynamic differential-reference time-to-digital converter [39], were incorporated to reduce energy consumption. Consequently, these SRAM-based CIMs achieved superior output accuracy, fast operation speeds, and high energy efficiency. Nevertheless, the limited capacity, volatility, and large leakage current impeded their deployment in intricate neural network architectures. As depicted in Table 3, a mass storage capacity of 2.25 MB was attained utilizing RRAM, leveraging its compact footprint enabled by the 1T1R structure [100]. On the other hand, MRAM achieved accelerated computational speeds compared to RRAM, with a latency of 5 ns for 1-bit input and 1–8-bit weight configurations [114]. Additionally, the nonvolatility of emerging memories allowed the complete power-down of unselected cells, leading to substantial reductions in standby power and thereby enhancing energy efficiency when compared to SRAM-based CIMs.

## 5. Conclusions

The rapid progress of edge computing has led to a growing need for emerging NVM technologies with low power consumption, high speed, and long-term durability. This paper explores the potential of four emerging NVMs in replacing conventional MCU memories and demonstrates their unique advantages. The discussion on NVM circuit design focuses on bitcell structures, read and write circuits, and macro structures, summarizing existing strategies for optimizing area, energy, and reliability. Moreover, previous works indicate that RRAM and MRAM offer notable benefits in CIM applications. Novel circuit designs, such as a binary interface structure and spintronic device integration, are effective in improving area and energy efficiency in CIM macros. As a future prospect, there is a need for universally applicable and energy-efficient strategies to leverage emerging NVMs in both MCUs and CIMs.

## Figures and Tables

**Figure 1 micromachines-15-00488-f001:**
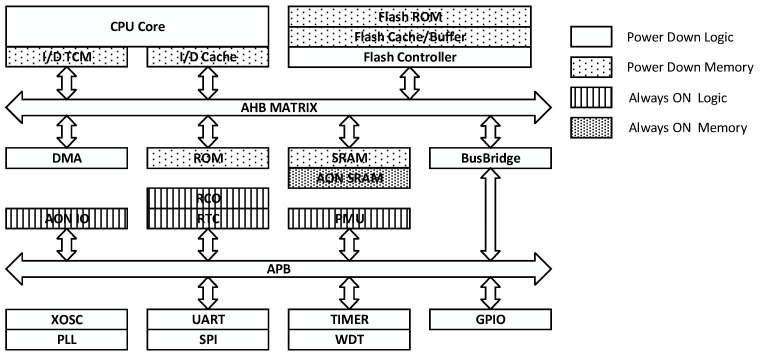
Block diagram of a prototypical MCU.

**Figure 2 micromachines-15-00488-f002:**
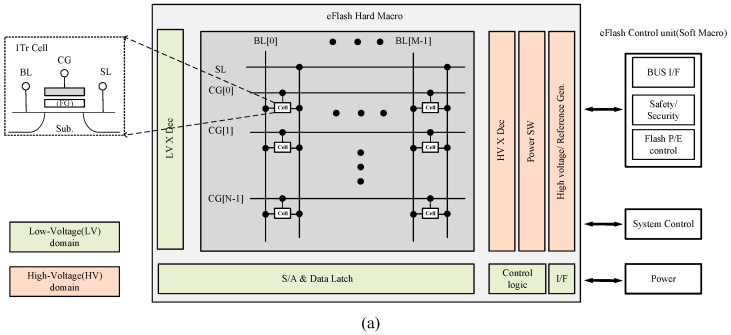
Memory floorplan of a typical (**a**) Flash and (**b**) SRAM embedded in MCUs.

**Figure 3 micromachines-15-00488-f003:**
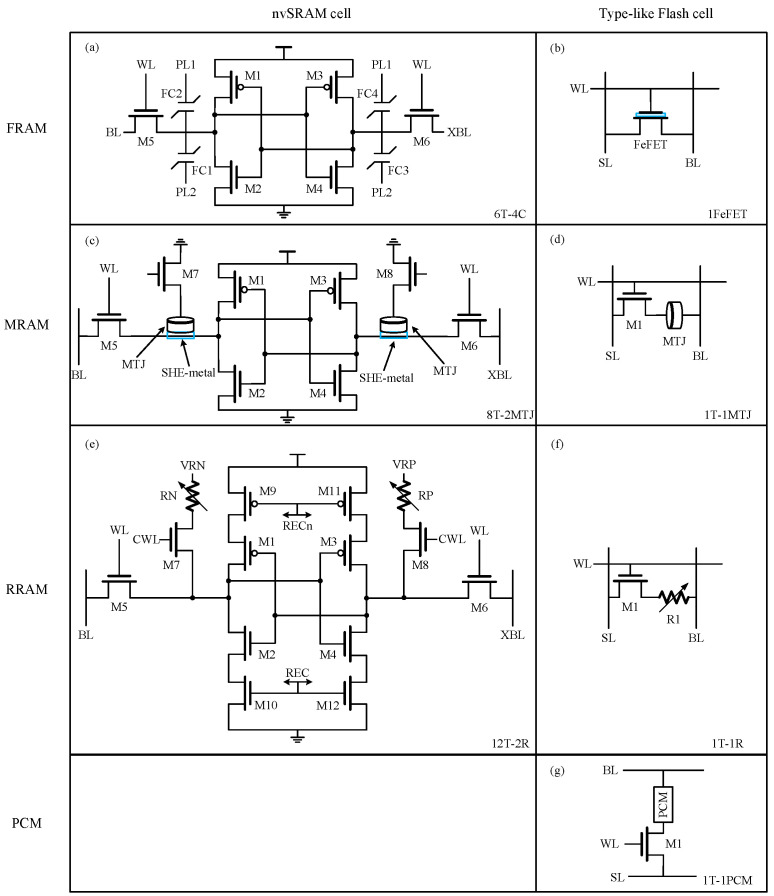
Bitcell structures of (**a**) a 6T-4C nvSRAM [20]; (**b**) a 1FeFET FRAM [72,73,74,75]; (**c**) a 8T-2MTJ nvSRAM [64]; (**d**) a 1T-1MTJ MRAM [76,77]; (**e**) a 12T-2R nvSRAM [19]; (**f**) a 1T-1R RRAM [21,54,78]; and (**g**) a 1T-1PCM PCM [22,79].

**Figure 4 micromachines-15-00488-f004:**
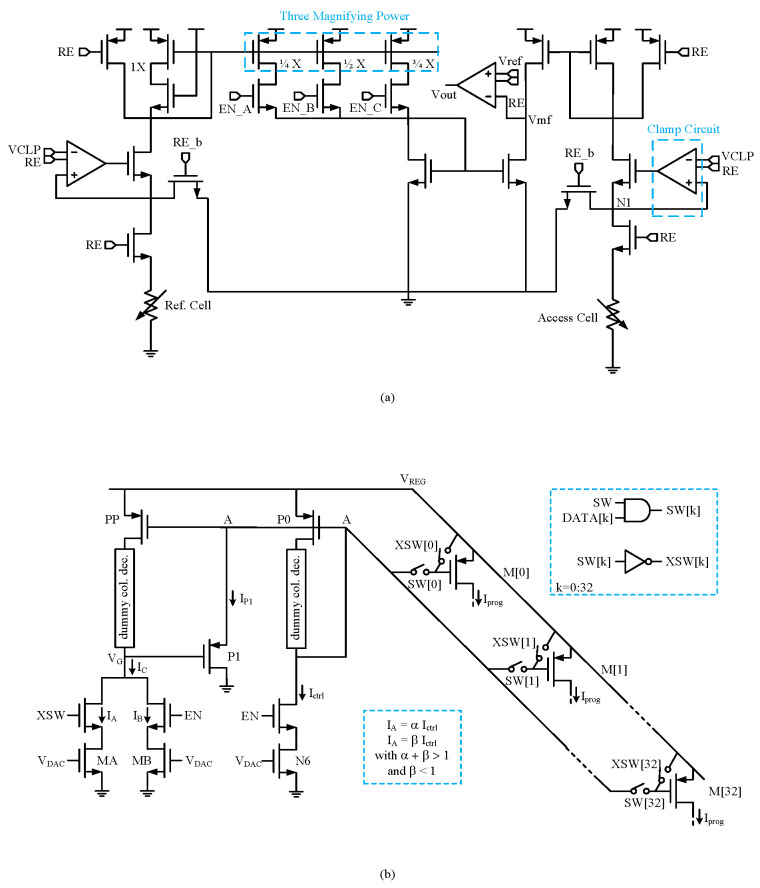
Circuit schematic of (**a**) a sense amplifier [54]; (**b**) a write driver [22].

**Figure 5 micromachines-15-00488-f005:**
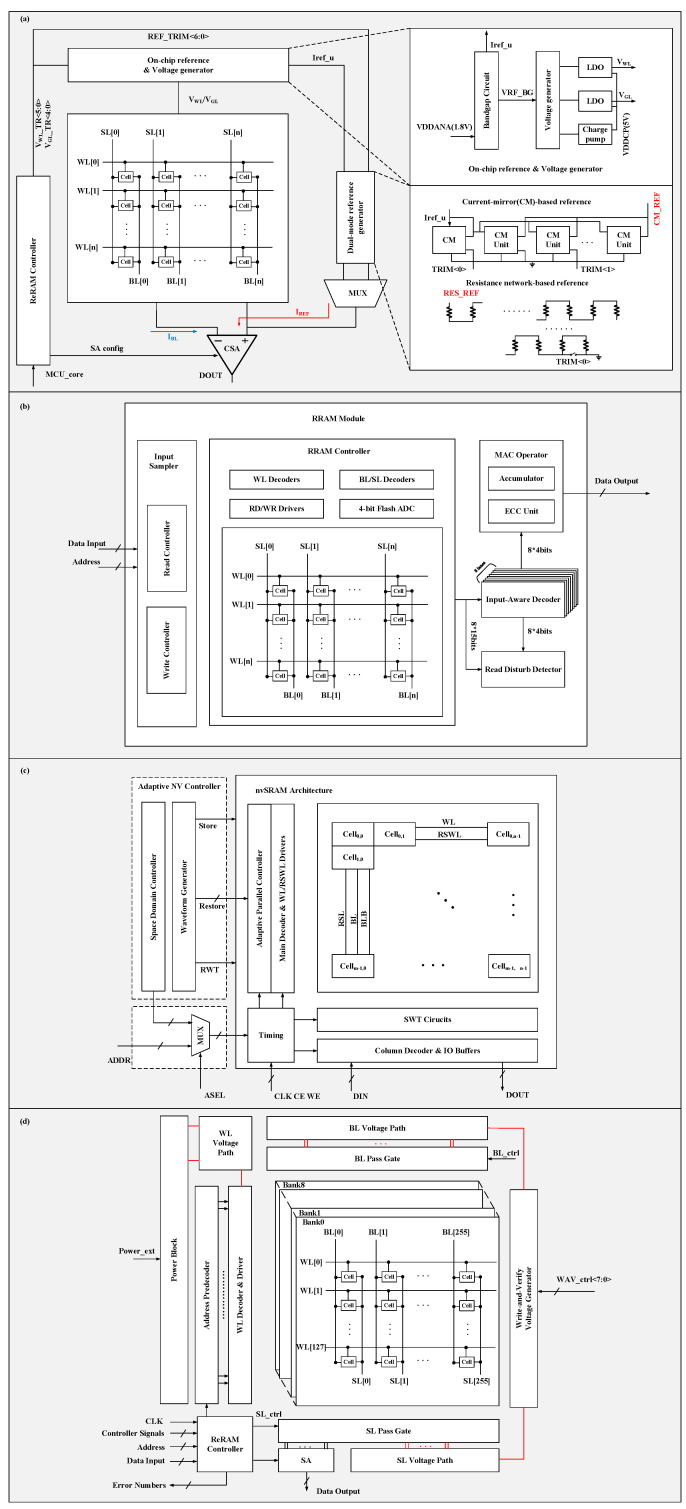
Structure diagrams of (**a**) a 1 MB RRAM macro [21]; (**b**) a 2.25 MB RRAM-based CIM macro [100]; (**c**) an adaptive RRAM-based nvSRAM macro [101]; and (**d**) a 256 KB RRAM macro [54].

**Figure 6 micromachines-15-00488-f006:**
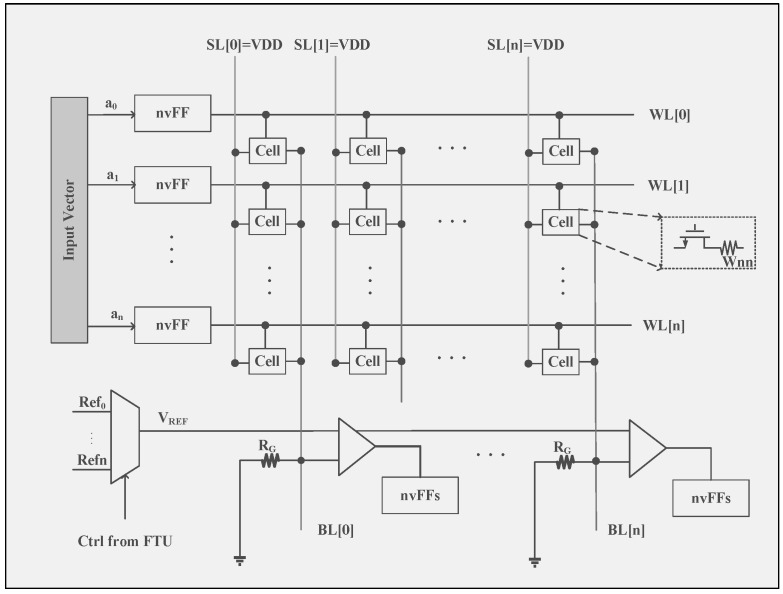
Architecture of an RRAM-based MVM engine for processing-in-memory [107].

**Figure 7 micromachines-15-00488-f007:**
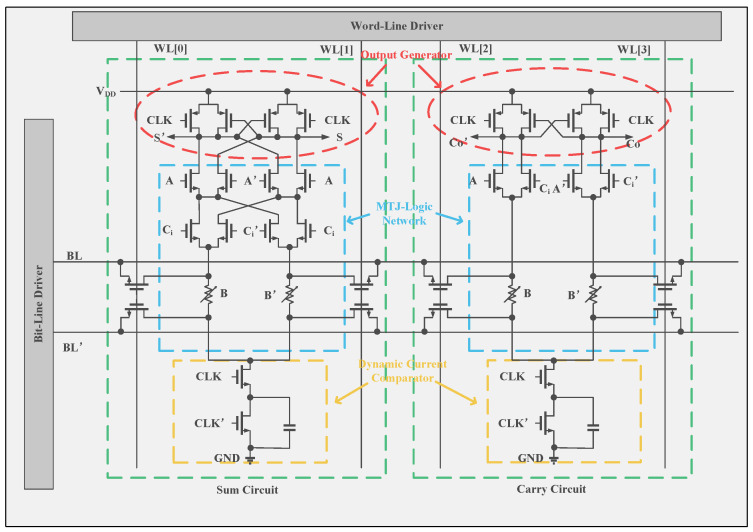
Circuit diagram of an MTJ-based nonvolatile full adder [61].

**Table 1 micromachines-15-00488-t001:** Comparison of eFlash and type-like Flash NVMs in MCUs.

Performance Metrics	JSSC’14 [46]	IEDM’16 [47]	ESSCIRC’13 [48]	IMW’19 [49]	ISSCC’13 [51]	ISSCC’16 [52]	JSSC’22 [53]	JETCAS’16 [54]	ISSCC’21 [55]	VLSI’22 [56]	VLSI’22 [57]	JSSC’18 [22]	VLSI’23 [58]
Architecture	/	/	/	/	Cortex-M0	8051 8 bit	RISC-V	/	RISC-V	Cortex-M0	Cortex-M33	/	/
Technology	40 nm CMOS	28 nm CMOS	65 nm CMOS	40 nm CMOS	130 nm CMOS, FRAM	65 nm CMOS, RRAM	40 nm CMOS, RRAM	180 nm CMOS, RRAM	22 nm FDSOI, MRAM	22 nm FDX, MRAM	22 nm CMOS, MRAM	110 nm BCD, PCM	28 nm FDSOI, PCM
Capacity	2 MB (code) 64 KB (data)	4 MB (code) 64 KB (data)	4 MB (code)	1 MB	64 KB	8 KB	2 MB	256 KB	4 MB	2 MB	2 MB	32 KB	21 MB
Cell structure	SG-MONOS	SG-MONOS	HS3P	eSTM	/	/	1T-1R	1T-1R	/	/	/	1T-1PCM	/
Cell size [F^2^]	/	0.053	/	0.049	/	/	0.64 *	20	/	/	/	0.7	0.019
Supply [V]	1.25	1.1	1.3	0.85–1.35	1.5	0.8	1.1	1.6/1.8	0.5–0.8	0.44–1.0	0.5–1.0	1.55–1.95	0.8
Active power [μW/MHz]	/	/	/	<150	112	33	135 mW	/	49.4 mW	387	158 mW	/	/
Standby power [μW]	/	/	/	<10	/	/	/	/	1.7	70	468	/	80 *
Max freq. [MHz]	160	200	81.5	/	8	>100	/	25	450	70	190	10	400
Endurance [cycles]	10 K (code) 10 M (data)	10 K (code) 1 M (data)	/	10 K	/	/	>${{3\times 10}^{5*}}^{5*}$	${{2\times 10}^{8}}^{8}$	/	/	/	$${10^{5}}^{5}$$	/
ECC	Yes	/	/	Yes	/	/	Yes	Yes	Yes	/	/	/	/

* It is estimated from figure in source.

**Table 2 micromachines-15-00488-t002:** Comparison of SRAM and nvSRAM in MCUs.

Performance Metrics	ISSCC’14 [66]	COOL CHIPS’20 [67]	A-SSCC’18 [68]	A-SSCC’19 [69]	BioCAS’14 [20]	TCAS-I’23 [19]	VLSI’18 [64]	TCAS-Ⅱ’21 [70]
Architecture	/	Cortex-M4	/	/	Cortex-M0	/	MSP430	/
Technology	65 nm CMOS	22 nm FDX	40 nm CMOS	110 nm CMOS	130 nm CMOS, FRAM	130 nm CMOS, RRAM	45 nm CMOS, MRAM	90 nm CMOS, RRAM
Non-volatility	N	N	N	N	Y	Y	Y	Y
Capacity	128 KB	256 KB	64 KB	2.5 MB	16 KB	64 KB	32 KB	/
Cell structure	6T	6T	6T	6T	6T-4C	12T-2R	8T-2MTJ	4T-2R
Cell size [F^2^]	2.159	0.26 *	2.888	1.84 *	/	16 *	/	2.83
Supply [V]	1.2	0.55	3.3	1.5	1.2	1.1–2.2	1.1/1.6	1.5
Active power [μW/MHz]	25	6.3 (MEP)	174 (Read) 180 (Write)	90 (Read) 105 (Write)	/	/	/	/
Standby power [μW]	/	6.6	0.33	0.73	/	/	2	/
Max freq. [MHz]	/	40	42	147	24	50	25	10 *

* It is estimated from figure in source.

**Table 3 micromachines-15-00488-t003:** Comparison of state-of-the-art CIM based on SRAM, RRAM, and MRAM.

Performance Metrics	ISSCC’20 [36]	ISSCC’21 [109]	ISSCC’21 [38]	ISSCC’22 [39]	ISSCC’20 [111]	ISSCC’21 [110]	ISSCC’22 [100]	VLSI’19 [112]	VLSI’20 [113]	ISCAS’23 [114]
Technology	28 nm CMOS	22 nm CMOS	28 nm CMOS	28 nm CMOS	22 nm CMOS, RRAM	40 nm CMOS, RRAM	40 nm CMOS, RRAM	22 nm CMOS, MRAM	22 nm CMOS, MRAM	28 nm CMOS, MRAM
Capacity	64 KB	64 KB	384 KB	1 MB	256 KB	8 KB	2.25 MB	/	/	/
Supply [V]	0.7–0.9	0.72	0.7–0.9	0.65–0.9	0.7–0.9	0.9	0.9	/	/	1
Input Precision [bit]	4/4/8	1–8	4/8	4/8	1–4	1–8	1–8	/	/	1–16
Weight Precision [bit]	4/8/8	4/8/12/16	4/8	4/8	2–4	1–8	1–8	/	1.7	1–8
Output Precision [bit]	12/16/20	16 (4b/4b) 24 (8b/8b)	12/20	14/22	6–11	20	32	4	4	8–16 (1b IN) 24–32 (1b IN)
Energy efficiency [TOPS/W]	47.85–68.44/ 23.26–33.52/ 11.54–16.63	24.7 (8/8/24b) 89 (4/4/16b)	60.28–94.31/ 15.02–22.75	84.45–112.6/ 21.19–27.75	121.38	56.67	60.64	9.2	19.6	25.43 (1/8/15b) 129.83 (1/1/8b)

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
