# Peer review of "A Survey of Emerging Memory in a Microcontroller Unit"

_micromachines, 2024, doi:10.3390/mi15040488_

Round 1

Reviewer 1 Report

Comments and Suggestions for Authors

This manuscript titled "A survey of emerging memory in microcontroller unit" is introducing the a review of the emerging memory application in embedded systems. The topic supposed to include FRAM, RRAM, MRAM and PCM for both nonvolatile memory applications and compute-in-memory applications. The topic itself is interesting, but lacks lots of details. Thus, the reviewer would suggest mandatory revision of the manuscript, before considering as a publication in Micromachines. Here provides some critical points for improvement.

1. The purpose of the paper is to apply emerging memories into real electronic systems, the original candidates are not comprehensively compared in each section.

2. Sections 2 and 3 is trying to compare SRAM and Flash (NOR) with the emerging memories. However, what the authors are showing in the manuscript for emerging memory designs are more DRAM like cells. Not sure if the authors are aware of what is what.

3. Section 3.1 title Bitcell design: cell size focus. Here the cell size term, is repeated with the Table 1 term cell size. Are they stands for same cell size ?

4. Section 3.1 showed several bitcell structures of nvSRAM and Type-like Flash cells. Are these standard/classical designs?

5. Figure 2 (c) M7 and M8 signal control not illustrated.

6. Section 4 circuit design for CIM based Emerging NVM only includes RRAM. Contents and title are mismatching.

7. Overall comparisons between current candidates and emerging memories are required. Especially for pros and cons.

Reviewer 2 Report

Comments and Suggestions for Authors

This paper focuses on reviewing latest developments in emerging nonvolatile memories with their peripheral designs and promising prospects for implementation in MCUs for future applications.

1. This survey demonstrates the requirements for NVM integrations for edge computing as the traditional power saving approaches like complete power shutdowns and energy autonomous systems take massive tradeoffs of efficiency. Then, four emerging NVMs are introduced as the substitute of conventional memories and each of them (FRAM, RRAM, MRAM, PCM) shows unique advantages over traditional memories in MCUs. Later, the discussions shift to the circuit design of NVM.  For bitcell design, the common transistor-NV device setup offering higher storage density, as some redundancy designs enhance the energy efficiency, reliability and flexibility for CIM. In terms of read and write peripheral circuit, this paper discussed various techniques, which provides flexibility of programming for maximizing energy efficiency and ensuring reliability. When considering macro structures, area efficiency is prioritized as power gating, adaptive parallel controllers and voltage generators are introduced. Last, the author provides a review of two examples of NVM-emerged CIM macros. With implementation different approaches and strategies, it reveals proper performance and energy efficiency advantages while some shortcomings are yet to be optimized.

2. The manuscript reviews the emerging NVMs for edge devices applications, while also extended the topic to the circuit design and CIM programming. However, the purpose of implementing certain technique is yet to be clear, such as the necessity for solving critical weakness and extra gains in NVM macro. The CIM part is also lacks of background introductions as what kind of tasks encountered.

3. The manuscript lacks of the authors perspective about the nonvolatile memories in MCU. The conclusion part is too long.

4. Comparison of different memory types can be found in several previous work. However, performance comparison of nonvolatile memory based MCU should also be surveyed and reported.

5. Recent IEDM 23, ISSCC 24 MCU paper should be highlighted to demonstrate the state-of-the-art.

6. The number of references is insufficient for a survey-type manuscript.

Round 2

Reviewer 1 Report

Comments and Suggestions for Authors

Thank the authors addressing the technical issues in the previous version manuscript. The reviewer has no more questions.

Please make sure to provide clear version of Table1 and Table2 in the final public version.